# Estimates and correlates of district-level maternal mortality ratio in India

**Srinivas Goli** [1,2]*, **Parul Puri** [1], **Pradeep S. Salve** [1], **Saseendran Pallikadavath**[3], **K. S. James** [1]

**1** International Institute for Population Sciences, Mumbai, Maharashtra, India, **2** University of Western Australia (UWA), Perth, Australia, **3** University of Portsmouth, Portsmouth, United Kingdom

* srinivasgoli@iipsindia.ac.in

**Data Availability Statement:** Data availability statement All data relevant to and included in the study are available in the public domain at the following links: HMIS: https://hmis.nhp.gov.in/#!/standardReports.

## Abstract

Despite the progress achieved, approximately one-quarter of all maternal deaths worldwide occur in India. Till now, India monitors maternal mortality in 18 out of its 36 provinces using information from the periodic sample registration system (SRS). The country does not have reliable routine information on maternal deaths for smaller states and districts. And, this has been a major hurdle in local-level health policy and planning to prevent avoidable maternal deaths. For the first time, using triangulation of routine records of maternal deaths under the Health Management Information System (HMIS), Census of India, and SRS, we provide Maternal Mortality Ratio (MMR) for all states and districts of India. Also, we examined socio-demographic and health care correlates of MMR using large-sample and robust statistical tools. The findings suggest that 70% of districts (448 out of 640 districts) in India have reported MMR above 70 deaths—a target set under Sustainable Development Goal-3. According to SRS, only Assam shows MMR of more than 200, while our assessment based on HMIS suggests that about 6-states (and two union territories) and 128-districts have MMR above 200. Thus, the findings highlight the presence of spatial heterogeneity in MMR across districts in the country, with spatial clustering of high MMR in North-eastern, Eastern, and Central regions and low MMR in the Southern and Western regions. Even the better-off states such as Kerala, Tamil Nadu, Andhra Pradesh, Karnataka, and Gujarat have districts of medium-to-high MMR. In order of their importance, fertility levels, the sex ratio at birth, health infrastructure, years of schooling, postnatal care, maternal age and nutrition, and poor economic status have emerged as the significant correlates of MMR. In conclusion, we show that HMIS is a reliable, cost-effective, and routine source of information for monitoring maternal mortality ratio in India and its states and districts.

## Introduction

Maternal mortality refers to death from any complications during pregnancy and childbirth or within 42 days of termination of pregnancy, irrespective of the duration and site of the pregnancy, but not from accidental or incidental causes [1]. Maternal Mortality Ratio (MMR) is the number of deaths per 100,000 live births. The recent global MMR estimates suggest significant progress. In particular, from 2000 to 2017, we notice a 38% decline in MMR—from 342

**Funding:** The authors received no specific funding for this work.

**Competing interests:** The authors have declared that no competing interests exist.

deaths to 211 deaths per 100,000 live births [2]. However, this average annual rate of reduction (2.2%) is less than the rate of decline needed (2.7%) to achieve the Sustainable Development Goal (SDG-3.1) of 70 maternal deaths per 100,000 live births by 2030 [3]. Though the improvement is remarkable, especially a steep decline in the absolute number of maternal deaths from 451,000 in 2000 to 295,000 in 2017, there are still 800 women dying each day due to pregnancy complications and childbirth worldwide. Sub-Saharan Africa and South Asia contribute about 86% of maternal deaths globally. In particular, South Asia accounts for 20% of maternal deaths, with 163 maternal deaths per 100,000 live births. Among South Asian countries, India is home to the highest maternal deaths (35000 maternal deaths) estimated globally in 2017. In percentage, the country accounts for 12% of global maternal deaths, next only to Nigeria (23%) [2].

According to the estimates of the Sample Registration System (SRS) of India, the MMR has significantly dropped from 400 per 100,000 live births in the early 1990s to 230 per 100,000 live births in 2008 and 130 in 2016 [4, 5]. Recent estimates of SRS have witnessed a steady decline in the MMR from 130 to 113 per 100,000 live births, with the highest rate in the State of Assam (215 per 100,000 live births) and the lowest in the state of Kerala (43 per 100,000 live births) [5]. The findings of previous studies [5–13] indicate that even though the overall MMR of India has drastically declined, the rate of decline in MMR is not uniform across the states. Some states have already achieved or are about to achieve the SDG goal of reducing the MMR to 70 per 100,000 live births by 2030. Nonetheless, seven out of eight Empowered Action Group (EAG) states, including Bihar, Madhya Pradesh, Chhattisgarh, Odisha, Rajasthan, Uttar Pradesh, and Uttarakhand, still have a long way to go to achieve the target set under SDG-3.1 [5]. EAG states contribute approximately 75% of India's total estimated maternal deaths, and Uttar Pradesh alone has more than 30% of the maternal deaths [5].

Moreover, owing to data limitations, previous studies [5–13] in India documented trends and patterns in MMR for only major states and 284 districts in nine EAG states, while the smaller states are completely excluded from the analyses. For a long time, the SRS has been the only reliable source of maternal mortality, which provides estimates for 18 major states [5]. However, SRS does not provide maternal mortality estimates for smaller states and districts. Although the Annual Health Survey (AHS) provided MMR estimates for 284 districts in nine EAG states from 2010 to 2013 [12], the survey was repealed after that, considering that National Family Health Survey (NFHS) would be redesigned to provide district-level health indicators for all Indian districts [14]. However, MMR estimates based on AHS never received as much prominence as SRS.

Moreover, considering within-state heterogeneity observed in other maternal and child health care indicators [15], we believe there must be considerable within-state variation in MMR. NFHS sample size does not allow deriving reliable estimates of maternal mortality at the state or the district level [15]. To our knowledge, there is not a single study in India that provides MMR estimates for the smaller states and all the districts.

On the other hand, earlier studies that investigated socioeconomic, demographic, and health care correlates of maternal mortality are either macro-level analyses based on the SRS data for 15 to 19 states or micro-level qualitative studies [6–13, 16–19]. The socioeconomic correlates identified based on the sample ranging from 15 to 18 states are less reliable, while micro-level local evidence is not nationally representative. Although many studies have documented clinical causes of maternal deaths [6, 8, 17–19], the identification of socioeconomic, demographic, and health care correlates immensely helps in designing policies and practices to avoid the death of women during pregnancy.

In the above context, this study makes two significant contributions: (1) for the first time, using Health Management Information System (HMIS) data, we provide MMR estimates for all 640 districts from 29 states and seven union territories of India. (2) Also, using the district-

level information from NFHS alongside HMIS, we have assessed socioeconomic, demographic, and health care correlates of MMR based on a significantly larger sample than previous studies. Also, for the first time, we have included district-level health infrastructure index and maternal health care variables as predictors of MMR.

## Methods

### Data input and processing

The study used data from multiple sources–HMIS (2017–18, 2018–19, 2019–20), the Sample Registration System (SRS, 2017–18), the Census of India (2011), and the National Family Health Survey (NFHS-4, 2015–16). HMIS is a web-based Monitoring Information System initiated by the Ministry of Health & Family Welfare, Government of India. Thus, HMIS data is the official data source for monitoring and evaluating various health programmes of the Government of India [20]. It provides India's consolidated public and private health facility-based service statistics data on the reproductive, maternal, neonatal, child, and adult health indicators. We have accessed the unit level data through the open-access link (https://hmis.nhp.gov.in/#!/standardReports) available in the public domain from the HMIS website.

An independent evaluation of completeness of HMIS records of maternal and child health indicators in 2016 suggests an average of 88.5% completeness, while it is as high as 94.6% for maternal health care indicators [21]. Moreover, HMIS has continuously improved its information recording system over the years. Thus, we would expect much better-quality information for 2018 to 2020 than what was observed in 2016.

The SRS has been a gold standard source for fertility and mortality data for more than five decades. It is the largest demographic and health survey in the country, which gives reliable estimates at the national and state level separately by urban and rural areas. The dual registration system, huge sample size, and verbal autopsy instruments make the estimates of SRS more reliable and representative at the national and state level (for details, see Office of the Registrar General of India, 2020) [5]. The NFHS is the largest sample survey that provides information on population, health, and nutrition for states and districts of India (for details see IIPS and ICF Macro, 2017) [15]. The population of women in the age group 15–49 years is drawn from the Census of India 2011 [22].

For the present study, we have analysed a total of 61,982,623 live births and 61,169 maternal deaths recorded in HMIS during 2017–19. HMIS enumerated numbers are considerably higher than the SRS sample of 429,173 live births and 525 maternal deaths at the all-India level during 2015–17. Further, the estimated annual number of births in India based on the birth rate from SRS has been about 25 million in recent years, which will amount to about 75 million in three years from 2017 to 2019 [5]. This suggests that HMIS covers nearly 77% of all live births in India, and such a high number can produce fairly reliable estimates despite potential coverage errors. This study is reported as per the Strengthening the Reporting of Observational Studies in Epidemiology (STROBE) checklist (S1 Table).

### Patient and public involvement

It was not appropriate or possible to involve patients or the general public in our research's design, conduct, reporting, or dissemination plans.

### Measures

Our outcome variable is the MMR estimated using live births and maternal deaths recorded through HMIS during 2017–19. Based on variables related to maternal deaths in the previous

literature [6, 8–14, 16–19, 23, 24] and considering data availability, we have included some key maternal health care, demographic and socioeconomic predictors to explain MMR variation across the districts of India. The predictor variables include health infrastructure index (HII) antenatal care, postnatal care, institutional delivery, mean age at first birth, contraception use in women, the mean number of children ever born, percentage of underweight, and anaemic women, years of schooling, household size, percentage of women in poor wealth status, and the sex ratio at birth. Detailed definitions and descriptions of the variables are mentioned in Tables 1 and 2, respectively.

**Table 1. Description of the study variables.**

| Variable | Definition | Data source |
|---|---|---|
| MMR | Death of women due to pregnancy or within 42 days of termination of pregnancy, irrespective of the duration and site of the pregnancy, from any cause related to or aggravated by the pregnancy or its management but not from accidental or incidental causes. Maternal Mortality Ratio (MMR) is measured as deaths for 100000 live births. | Authors estimation from HMIS |
| HII | Multidimensional measure calculated using information collected for rural health infrastructures on several items: number of district hospitals, Community Health Centers (CHCs), Primary Health Centers (PHCs), Sub-Centers (SCs), Doctors, Nurses, Auxiliary Nurse Midwife (ANM), Accredited Social Health Activist (ASHA), Anganwadi Worker (AWW) per 1000 population. We used the model of HDI for estimating dimension-free numbers then aggregated them to generate Health Infrastructure Index (HII). The HII is adjusted for the share of the urban population in ordered to give weightage to urban health infrastructure (especially private health infrastructure). Weight is equivalent to the share of the urban population in the district. | Author's estimation from Rural Health Statistics reports of India. |
| 4 or more ANCs | Percentage of women who received four or more antenatal care services. | Authors estimation from NFHS (2015–16) |
| PNCs | Percentage of women who received postnatal care within 48 hours. | Authors estimation from NFHS (2015–16) |
| Institutional delivery | Percentage of women delivered a child in hospital settings. | Authors estimation from NFHS (2015–16) |
| Contraception | Percentage of women currently using any modern method of contraception | Authors estimation from NFHS (2015–16) |
| Body mass index | Body Mass Index is the height for weight score of adult women in the age group 15–49 years. | Authors estimation from NFHS (2015–16) |
| Anaemic | Haemoglobin levels below (<12 mg/dl for non-pregnant and <11 mg/dl for pregnant) are considered anaemic. | Authors estimation from NFHS (2015–16) |
| Mean age at first marriage | Age at first marriage as reported by women in years | Authors estimation from NFHS (2015–16) |
| Mean age at first birth | Age at first birth as reported by women in years | Authors estimation from NFHS (2015–16) |
| Sex ratio at birth | Number of girls per 1000 boys at the time of birth | Authors estimation from NFHS (2015–16) |
| Mean children ever born (CEB) | Mean number of children ever born per woman | Authors estimation from NFHS (2015–16) |
| 10 or more years of schooling | Percentage of women who have completed 10 years or more schooling. | Authors estimation from NFHS (2015–16) |
| Average household size | The average number of persons living in a household | Authors estimation from NFHS (2015–16) |
| Urban Population | Share of the urban population in a district | Authors estimation from NFHS (2015–16) |
| Poor household economic status | Share of poor households derived from the wealth index. The wealth index is derived by assigning scores based on the number and kinds of consumer goods they own, ranging from a television to a bicycle or car, and housing characteristics such as the source of drinking water, toilet facilities, and flooring materials. These scores are derived using principal component analysis. National wealth quintiles are compiled by assigning the household score to each usual (de jure) household member, ranking each person in the household population by their score, and then dividing the distribution into five equal categories, each with 20 percent of the population. We have considered the first two quintiles as relatively poor households [17]. | Authors estimation from NFHS (2015–16) |

**Table 2. Descriptive statistics of the study variables.**

| Variable | Observations (No. Districts) | Mean | Standard Deviation | Minimum | Maximum |
|---|---|---|---|---|---|
| Maternal Mortality Ratio | 639 | 142.21 | 127.84 | 0 | 1671 |
| HII | 640 | 0.540 | 0.37 | 0.09 | 5.55 |
| 4 or more ANCs (%) | 640 | 52.46 | 26.01 | 0.85 | 99.14 |
| PNCs within 48 hours of delivery (%) | 640 | 62.71 | 17.70 | 0. | 100. |
| Institutional delivery (%) | 640 | 80.34 | 16.61 | 10.25 | 100 |
| Contraception (%) | 640 | 50.84 | 17.16 | 2.73 | 84.81 |
| Body Mass Index | 640 | 17.62 | 8.76 | 1.17 | 45.06 |
| Anaemic (%) | 640 | 51.58 | 12.09 | 13.85 | 82.77 |
| Mean age at Marriage | 640 | 18.62 | 1.36 | 15.64 | 23.38 |
| Mean age at first birth | 640 | 20.60 | 1.02 | 18.24 | 24.99 |
| Mean Children Ever Born | 640 | 2.46 | 0.44 | 1.57 | 3.82 |
| 10 or more years of schooling (%) | 640 | 28.13 | 14.26 | 5.6 | 86.47 |
| Sex Ratio at Birth | 626 | 925.11 | 110.86 | 600 | 1537 |
| Average household size | 640 | 5.68 | 0.76 | 3.98 | 8.45 |
| Scheduled Castes/Tribes Population (%) | 640 | 38.16 | 23.27 | 0.70 | 100 |
| Urban Population (%) | 640 | 27.33 | 21.66 | 0 | 100 |
| Poor households (%) | 640 | 40.65 | 25.73 | 0.12 | 90.55 |

Note: ANCs–Antenatal care services, PNCs–Postnatal care services

## Estimation of Maternal Mortality Ratio (MMR)

We used the standard method for MMR computation using the information on the number of maternal deaths and live births reported in HMIS. Thus, mathematically it takes the following form:

$$\text{MMR}_{\text{district}} = \frac{\text{No.of maternal deaths in a district}}{\text{No. of live births in a district}} * 100000 \qquad (1)$$

However, considering that HMIS is a vital registration system that is prone to some coverage errors, we have attempted to standardise MMR estimates from HMIS for states and districts using gold standard information on maternal deaths from SRS and female population figures from the Census of India. Specifically, we used triangulation of data from the HMIS, SRS, and Census of India (2011) to derive the final MMR estimates. A calibration factor (Cf) was computed and used to account for the under-(over)-reporting of maternal deaths in HMIS by states and districts of India. The calibration factor was initially estimated for states as the ratio of MMR from SRS and HMIS, as shown in Eq (1). For the states where MMR estimates were missing in SRS, we used the estimates of Infant Mortality Rates (IMR) as a proxy for MMR estimates to compute the calibration factor. In this case, the calibration factor was the ratio of IMR from SRS and HMIS, as shown in Eq (2). The mathematical expressions for the aforementioned computations are as follows:

$$\text{Cf} = \frac{\text{SRS\_MMR}^{\text{State\_Estimate}}}{\text{HMIS\_MMR}^{\text{State\_Estimate}}} \qquad (2)$$

And, for the states where MMR is missing in SRS, we utilized the value of IMR as a proxy. In this case, the expression for computation of Cf can be written as follows:

$$Cf = \frac{SRS\_IMR^{State\_Estimate}}{HMIS\_IMR^{State\_Estimate}} \tag{3}$$

Then, we have adjusted the district estimates of each state using the calibration factor (Cf) derived for that particular state using the aforementioned procedure in Eqs 1 and 2. The adjusted MMR for each district was derived as below:

$$District\ MMR^{Adjusted} = District\ MMR^{Unadjusted} * Cf \tag{4}$$

Finally, we have derived the adjusted state estimates using adjusted district MMRs and district population weights. Population weight for each district is derived using the information on women 15–49 years of age from the Census of India, 2011. This procedure will adjust for district-level unequal size in error margins proportionately weighted by population size while deriving the state-level adjusted MMRs using HMIS data. The estimated MMR for each state is as follows:

$$State\ MMR^{Adjusted} = \frac{\sum_{i=1}^{n} District\ MMR^{Adjusted} * pw}{n} \tag{5}$$

Where pw is population weight defined as:

$$pw = \frac{Total\ female\ population\ of\ the\ district\ in\ age\ 15 - 49\ years}{Total\ female\ population\ of\ the\ state\ in\ age\ 15 - 49\ years} \tag{6}$$

## Geographical distribution and spatial clustering

We have carried out statistical analyses in three stages: First, we used GIS mapping to show the geographical distribution of MMR across the states and districts of India. In the second stage, to assess the extent of geographical clustering, univariate local Moran's I and Local Indicator of Spatial Association (LISA) cluster, and significance maps were employed. Spatial proximity was quantified using the Queen Contiguity matrix, which includes neighbours sharing geographical boundaries of non-zero length [24]. Moran's I statistics range between -1 and +1, where a positive, negative, and zero value indicates positive, negative, and no spatial autocorrelation, respectively [25, 26]. Our Cluster map depicts the locations (districts) with a significant local Moran's I statistic classified by spatial auto-correlation type; the color red symbolises the hot spots (districts with high MMR levels, with similar neighbours), green symbolises the cold spots (districts with low MMR levels, with similar neighbours). The light blue and pink color symbolises the spatial outliers (districts with high MMR levels but with low- MMR level neighbours and vice-versa).

## Ordinary least square regression model: Macro-level correlates

In the last stage, we have carried out an Ordinary least square (OLS) log-linear regression model to understand the maternal health care, demographic and socioeconomic correlates of MMR. We have modelled univariate (unadjusted) regression estimates in model 1. While from models 2 to 7, we have run six OLS regressions to avoid the collinearity between the explanatory variables. We have avoided highly collinear variables (r>0.60) in the same model based on the correlation matrix of the study explanatory variables.

The mathematical expression of the model is given below:

$$Y_{(Log\_MMR)} = a + b_1 X_{1(Log\_HII)} + b_2 X_{2(Log\_No.of\ ANCs)} + b_k X_{k...} + \varepsilon_i \tag{7}$$

Where Y is the outcome variable (*i.e.* MMR), which is influenced by a set of predictor variables $X_1, X_2, X_3$---------$X_K$ (*e.g.*, HII, antenatal care, postnatal care, institutional delivery, mean age at first birth) in the manner specified with parameters $\beta_1, \beta_2$.............$\beta_K$.

Statistical analyses were performed using STATA 16 statistical software (Stata Corporation, College Station, TX, USA).

## Results

### Geographical variation and spatial clustering of maternal mortality

Fig 1 depicts the spatial pattern of MMR across 29 states and 7 union territories in India. Findings underline considerable geographical heterogeneity in MMR across Indian states. MMR was categorised into four groups, less than 70, 70–139, 140–209, greater than or equal to 210 deaths per 100000 live births. The first cut-off was taken at 70, which is a primary target under SDG-3 for MMR, while the second cut-off at 140 is a second target under SDGs. Further, the same interval has been taken to create two more categories [15]. Such categorisation allows classifying Indian states and districts as those achieved, near to achieve, or far from the achievable SDG target 3.1.

Among the states, the highest MMR is found in Arunachal Pradesh (284) and the lowest in Maharashtra (40). The findings illustrate five states, including Arunachal Pradesh (284), Manipur (282), Andaman and Nicobar Island (275), Meghalaya (266), and Sikkim (228), have MMR greater than or equal to 210. Nine states and two union territories have MMR in the range of 140–209. These states are Nagaland (143), Punjab (143), Chhattisgarh (144), Jammu and Kashmir (151), Delhi (162), Rajasthan (162), Bihar (164), Madhya Pradesh (179), Lakshadweep (208), Uttar Pradesh (208), and Assam (209).

Eleven states have MMR in the range of 70–139: Gujarat (76), Jharkhand (78), Karnataka (85), Haryana (90), Goa (91), West Bengal (100), Uttarakhand (107), Tripura (119), Himachal Pradesh (127), Mizoram (131), and Odisha (138). Furthermore, the estimates indicate that nine out of 36 provinces have MMR less than 70: Chandigarh (15), Maharashtra (40), Puducherry (41), Kerala (44), Daman and Diu (48), Telangana (53), Tamil Nadu (56), Dadra and Nagar Haveli (61) and Andhra Pradesh (64) (S2 Table).

Fig 2 depicts the geographical pattern of MMR in 640 districts of India. Among the districts, the highest MMR is found in Tirap district in Arunachal Pradesh (1671), while thirteen districts reported the lowest MMR levels, these included seven districts from Arunachal Pradesh, two districts from Himachal Pradesh and one district from Jammu & Kashmir, Maharashtra, Puducherry, and Uttarakhand, each. The results indicate that 192 districts have MMR less than 70 and 210 districts fall in the range of 70–139. However, about 124 districts have MMR in the range 140–209, and 114 districts fall in the category of greater than or equal to 210. In particular, among the districts with MMR greater than or equal to 210, 46 districts belonged to the Central Region, and 33 districts are located in the North-eastern region; while 18 districts belonged to the Northern region and 16 to the Eastern region.

A majority of the districts in southern India and Maharashtra have an MMR of less than 70. 68 districts in Southern India have MMR less than 70, followed by Western (46 districts), Eastern (30 districts), and Northern (30 districts) regions. At the same time, North-eastern and Central regions have the least number of districts (12 and 6 districts respectively) that achieved the primary SDG target of MMR (S3 Table).

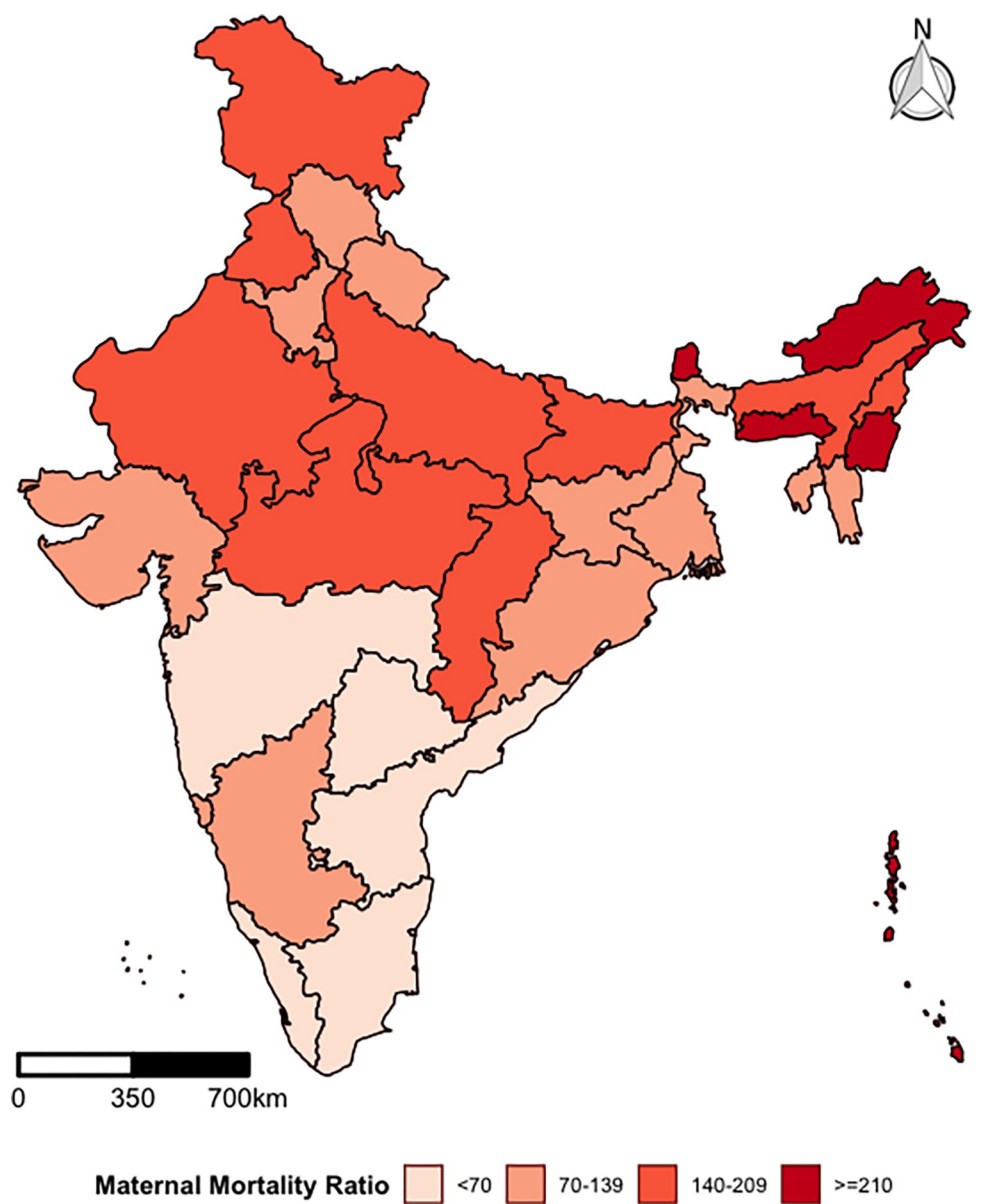

**Fig 1. Geographical pattern of maternal mortality ratio by states/union territories in India, HMIS.** (https://github.
com/datameet/maps/). (Authors generated the estimates employing the HMIS data).

However, Fig 2 also demonstrates the presence of huge within-state inequalities. For
instance, the state of Karnataka as a whole fall in the category of 70–139, but several of its dis-
tricts have an MMR above 140. Similarly, some districts in Tamil Nadu, Kerala, Andhra Pra-
desh, and Telangana also have MMR above 140, despite all four states falling in the category of
MMR below 70 at the state level. A similar kind of district-level heterogeneity is observed in
other states as well.

Supporting these findings, the results from univariate LISA (Fig 3A and 3B) also suggest the
presence of spatial heterogeneity in MMR with statistically significant spatial autocorrelation
(Moran's I = 0.229, p-value = 0.001) across districts in the country. Geographical clustering of
high MMR was observed in the North-eastern and parts of the Central region. Southern and
Western regions in the country reported a noticeable geographical clustering of low MMR.

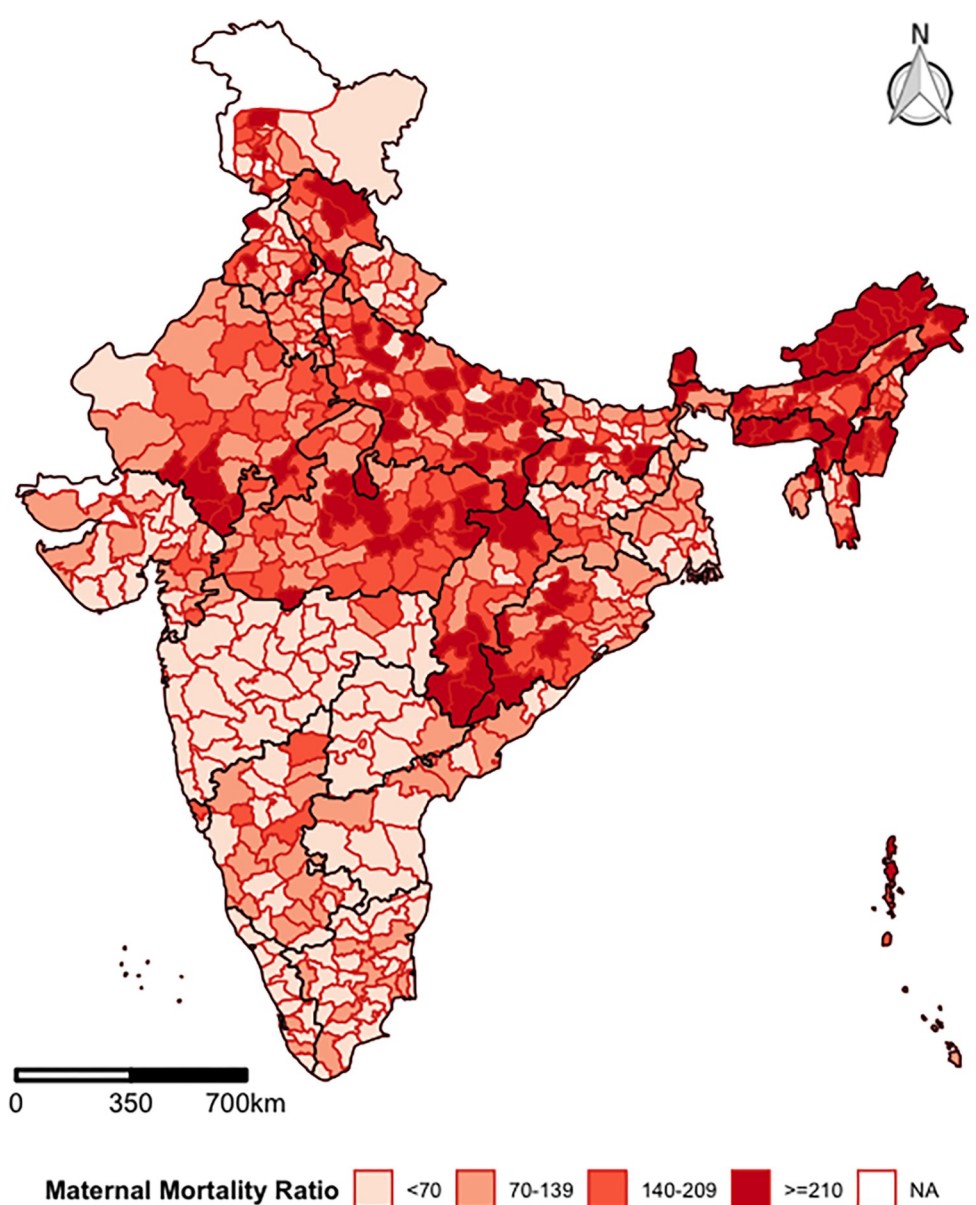

**Fig 2. Geographical pattern of maternal mortality ratio by 640 districts in India, HMIS.** (https://github.com/datameet/maps/). (Authors generated the estimates employing the HMIS data).

Furthermore, bivariate LISA assessed the spatial association between the selected background variables and MMR for 640 districts in the country. The findings from the bivariate spatial analysis are presented in S1 Fig. Bivariate analysis suggests that regions with low age at first birth, low contraception use, a high mean number of children ever born, higher percentage of underweight and anaemic women are more likely to report higher MMR. Also, the MMR is found to be higher for the districts with a lower percentage of four or more ANC, lower percentage of postnatal care, lower percentage of institutional deliveries, and lower health infrastructure. Districts with a lower percentage of ten or more years of schooling, a larger mean household size, a low rate of the urban population, and a higher percentage of the poor economic status population are more likely to report higher MMR. However, there are

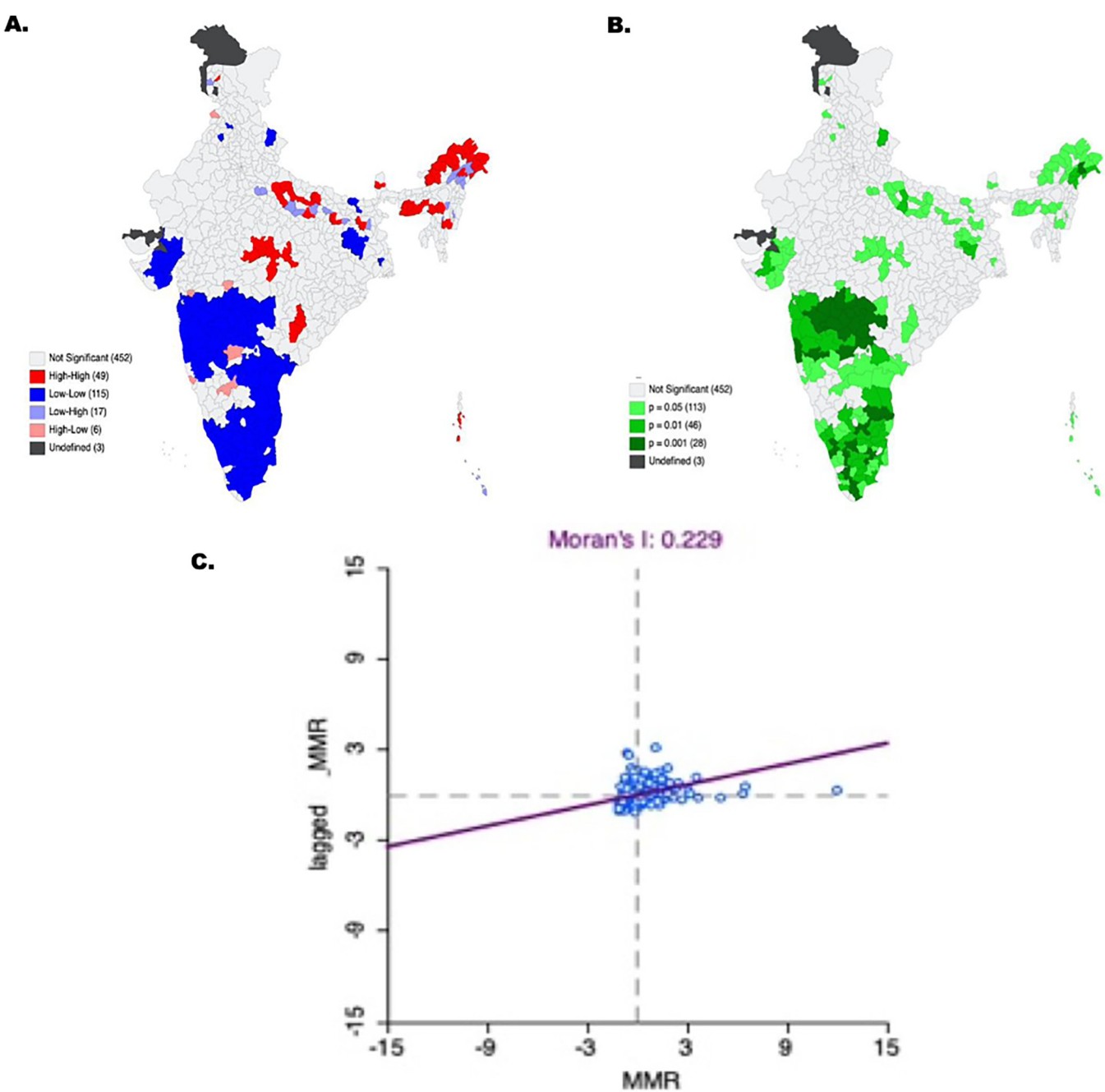

**Fig 3. Univariate Moran's I for maternal mortality ration in India.** (https://github.com/datameet/maps/). (Authors generated the estimates employing the HMIS data). 3A. (https://github.com/datameet/maps/). (Authors generated the estimates employing the HMIS data). 3B. (https://github.com/datameet/maps/). (Authors generated the estimates employing the HMIS data).

several exceptional cases found. Districts with higher age at first birth and lower prevalence of anaemic women are also home to higher MMR, thus indicating spatial heterogeneity in the relationship between MMR and socioeconomic characteristics. It also suggests that a multitude of factors influences MMR. The following section investigates the net effect of socioeconomic correlates after controlling for confounders.

## Factors associated with maternal mortality: A macro-level analysis

Table 3 presents the net effect of socioeconomic, demographic, and health care correlates of maternal mortality ratio based on the OLS regression model. The univariate regression results

**Table 3. Log-linear regression estimates: Correlates of maternal mortality ratio in India.**

| VARIABLES | Univariate | Multivariable | | | | | |
|---|---|---|---|---|---|---|---|
| | (1) | (2) | (3) | (4) | (5) | (6) | (7) |
| | Model 1 | Model 2 | Model 3 | Model 4 | Model 5 | Model 6 | Model 7 |
| HII | -0.542*** | | -0.551*** | | | -0.494*** | -0.482*** |
| | (0.092) | | (0.106) | | | (0.154) | (0.149) |
| 4 or more ANCs | -0.223** | -0.273*** | -0.0477 | | | 0.181 | |
| | (0.072) | (0.102) | (0.109) | | | (0.134) | |
| PNCs within 48 hours of delivery | -0.274* | -0.211 | -0.279* | | | -0.379** | |
| | (0.142) | (0.162) | (0.159) | | | (0.157) | |
| Institutional delivery | -0.311 | 0.323 | 0.316 | | | 0.386 | 0.540 |
| | (0.192) | (0.299) | (0.293) | | | (0.324) | (0.240) |
| Age at marriage | -1.058 | | | -1.662 | | -1.241 | |
| | (0.708) | | | (2.346) | | (2.767) | |
| Age at first birth | 0.018 | | | 7.905** | | 8.853** | 7.946*** |
| | (1.041) | | | (3.094) | | (3.537) | (1.217) |
| Contraception use | -0.106 | | | 0.219* | | 0.152 | |
| | (0.109) | | | (0.120) | | (0.132) | |
| Children ever born | 0.411*** | | | 1.822*** | | 1.944*** | 1.437*** |
| | (0.081) | | | (0.332) | | (0.444) | (0.413) |
| Body mass index | 0.573*** | | | 0.437*** | | 0.406*** | 0.324*** |
| | (0.097) | | | (0.113) | | (0.134) | (0.116) |
| Anaemic | 0.396** | | | 0.223 | | 0.303 | 0.325* |
| | (0.192) | | | (0.204) | | (0.200) | (0.194) |
| 10 or more years of schooling | -0.570*** | | | | | | -0.404*** |
| | (0.097) | | | | | | (0.145) |
| Sex ratio at birth | -0.960** | | | | -1.218*** | -1.078*** | -1.074*** |
| | (0.437) | | | | (0.416) | (0.401) | (0.393) |
| Average household size | 2.252*** | | | | | | |
| | (0.370) | | | | | | |
| SC/ST population | 0.009 | | | | -0.0144 | 0.188** | 0.188** |
| | (0.086) | | | | (0.0866) | (0.0933) | (0.087) |
| Urban population | -0.100 | | | | 0.0412 | 0.287*** | 0.310*** |
| | (0.061) | | | | (0.0767) | (0.0930) | (0.090) |
| Poor household economic status | 0.170*** | | | | 0.215*** | 0.0195 | |
| | (0.044) | | | | (0.0565) | (0.0810) | |
| District dummy | | | | | | Yes | Yes |
| Constant | | 4.987*** | 4.013*** | -19.06*** | 12.03*** | -18.74*** | -18.63*** |
| | | (1.010) | (1.008) | (4.442) | (2.868) | (5.471) | (4.887) |
| Observations | 638 | 638 | 638 | 640 | 623 | 621 | 623 |
| R-squared | | 0.018 | 0.058 | 0.110 | 0.042 | 0.176 | 0.158 |

Note: ANC–Antenatal care, PNC–Postnatal care, SC/ST- Scheduled Caste and Tribes; Robust standard errors in parentheses

*** p<0.01

** p<0.05

* p<0.1

in model 1 suggest that though all the coefficients are not statistically significant, but MMR shows an expected relationship with maternal health care, demographic and socioeconomic factors. In model 2, before controlling for other correlates, ANCs ($\beta$ = -0.273, $p<0.01$) is negatively and significantly associated with MMR; however, when we controlled for all other correlates in models 2, 5, and 6, 4 or more ANC visits did not show the desired relationship with MMR. Similarly, when we run the regression model considering only health infrastructure and maternal health care variables, health infrastructure ($\beta$ = -0.551, $p<0.01$) and PNCs within 48 hours of delivery ($\beta$ = -0.279, $p<0.1$) are negatively associated and statistically significant. Surprisingly, institutional delivery is positively associated but statistically insignificant across all the models.

Using only demographic variables, the results in model 4 suggest that age at first birth ($\beta$ = 7.905, $p<0.1$), ever use of contraception ($\beta$ = 0.219, $p<0.05$), and children ever born ($\beta$ = 1.822, $p<0.01$) are positively associated, while body mass index ($\beta$ = -0.437, $p<0.05$) is negatively associated with MMR. Model 5, which uses only socioeconomic variables, reveals that the sex ratio at birth ($\beta$ = -1.218, $p<0.01$) is negatively associated, while the poor economic status of the households ($\beta$ = 0.215, $p<0.01$) is positively linked to MMR. The SC/ST population share is positively associated ($\beta$ = 0.188, $p<0.05$) with MMR in Model 6. Model 7 controls for all variables suggests that health infrastructure ($\beta$ = -0.535, $p<0.01$), PNCs within 48 hours of delivery ($\beta$ = -0.370, $p<0.05$), body mass index ($\beta$ = -0.357, $p<0.01$) and year of schooling ($\beta$ = -0.437, $p<0.01$) are negatively associated, while age at first birth ($\beta$ = 7.431, $p<0.05$) and children ever born ($\beta$ = 1.589, $p<0.01$) are positively and significantly correlated with MMR. The institutional deliveries continue to show statistically insignificant negative relationships.

## Robustness checks: Data reliability assessment

The first robustness check parameter used in this study is the estimation of completeness of birth registration in HMIS. The estimated annual number of births in India is about 81 million in three years from 2017 to 2019, while reported cumulative live births during 2017–19 under HMIS is 62 million—this suggests that HMIS has coverage of 77% of all estimated live births in the country. Among major states, with 95%, Telangana and Kerala show the highest completeness of birth registration, while the corresponding figure is lowest in Uttar Pradesh (62%). However, 26 out of 37 states and union territories have complete birth registration equal to or above the national average. Twenty out of 37 states and 17 of 37 states show above 80% and 85% of birth registration completeness, indicating that HMIS information is highly reliable for deriving basic demographic estimates (Fig 4). Although the missing deaths or deaths that physicians were unable to code cannot be ignored, given their low proportion, conservatively, it is safe to assume that they did not affect the general regional pattern of MMR shown in this study.

Next, we compared HMIS based MMR estimates with corresponding estimates from SRS for the major states. At all India levels, SRS shows 130 in 2014–16 and 113 in 2016–18, while HMIS based estimate is 122 in 2017–19 (S2 Table). In Fig 5, we plot MMR estimates from SRS and HMIS. The MMR estimates from HMIS are close to SRS in socio-demographically better-off states (Andhra Pradesh, Gujarat, Karnataka, Tamil Nadu, Kerala, Maharashtra, etc.). The gap is slightly higher in socio-demographically weaker states (Assam, Bihar, Chhattisgarh, Uttar Pradesh, Madhya Pradesh, and Odisha). Despite a slight gap in MMR estimates from HMIS and SRS in a few states, the pattern remains more or less the same in the estimates from both sources: the MMR is higher in socio-demographically weaker states compared to their counterparts in socio-economically advanced states. Similar evidence can also be observed in the case of comparison of IMR from SRS and MMR from HMIS. We found a high positive

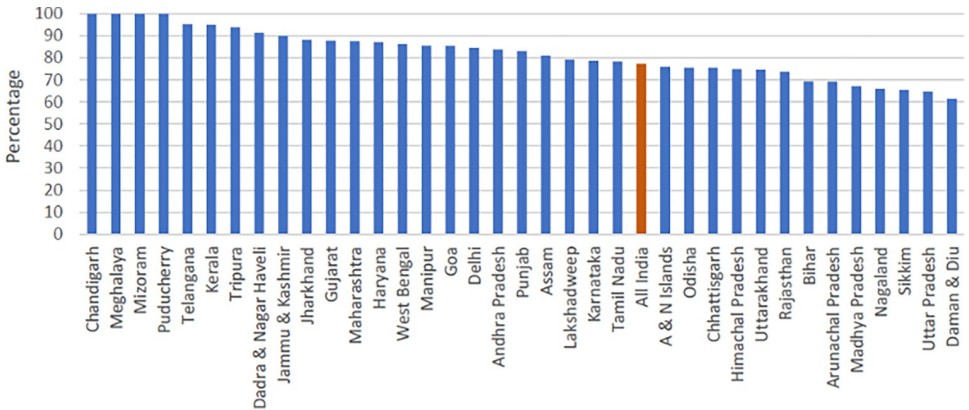

**Fig 4. Percentage of reported live births out of estimated live births by states in HMIS, 2017–19.**

correlation between IMR from SRS and MMR from HMIS with a correlation coefficient of 0.78 (Fig 6).

Thirdly, a comparison of MMR estimates from other sources with our estimates suggests that both SRS and HMIS based MMR is much lower than the Global Burden of Disease study estimate of 247·6 for 2015 but closer to estimates (145 in 2017) by WHO, UNICEF, UNFPA, World Bank Group and the United Nations Population Division [2, 5, 27]. Overall, our MMR estimates using HMIS more or less align with SRS estimates and the estimates from WHO, UNICEF, UNFPA, World Bank Group, and the United Nations Population Division [2, 5].

Fourth, we further compared a few other basic demographic estimates from HMIS (2017–19) with SRS (2018). For instance, IMR from HMIS (2017–19) is 26.2 against 32 from SRS (2018). Similarly, the Sex Ratio at Birth from HMIS (2017–19) is 108 against 111 from SRS (2018). While the Crude Birth rate in HMIS is 24, it is 20.2 in SRS. HMIS based IMR, SRB, and CBR estimates are also close to corresponding year estimates from the technical group's report on population projections [28] (S4 Table).

Fifth, the macro-level regression estimates show the expected direction of association between health infrastructure, maternal health care, and socio-demographic indicators. MMR also strengthens our belief that the estimates are in line with the status of districts' socio-

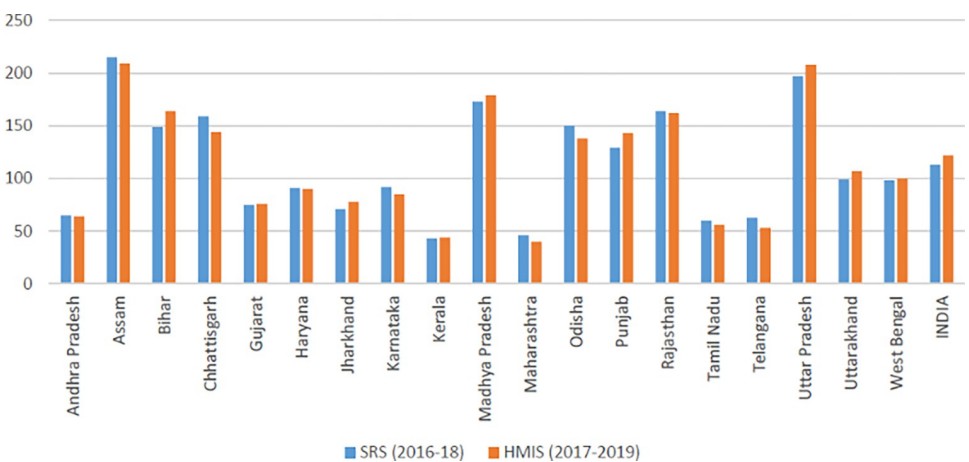

**Fig 5. Correspondence between MMR estimates from SRS and HMIS.**

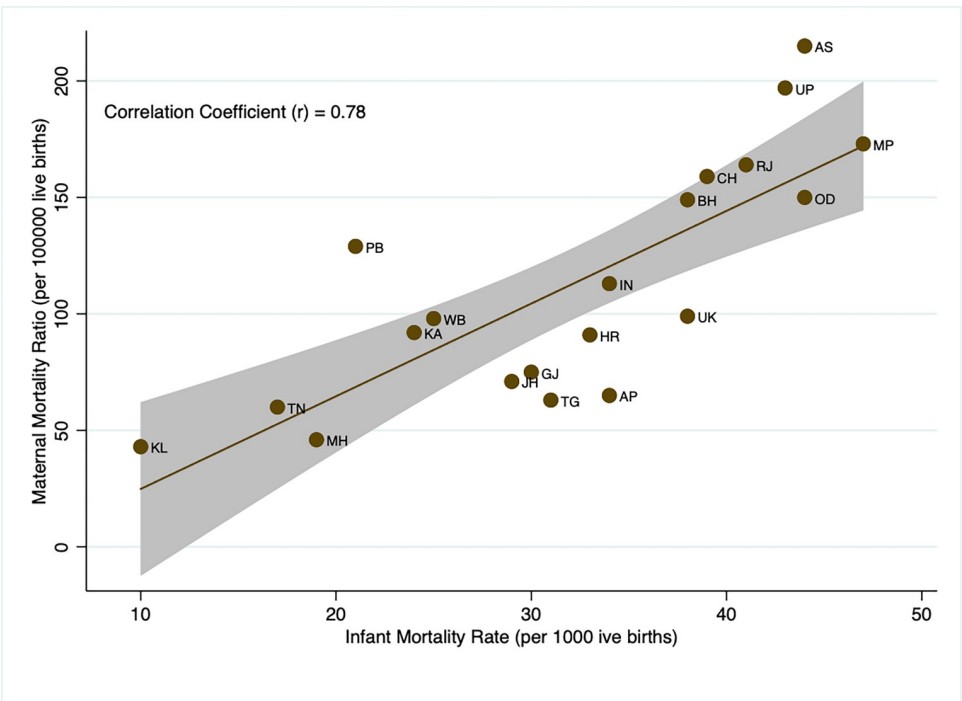

**Fig 6. Correlation between IMR estimates from SRS and MMR estimates from HMIS.**

demographic and health status. If there was a health facility-led bias in MMR registration, we would not have seen the expected direction of the relationship between these variables. Overall assessment of the quality of data reported in HMIS vis-à-vis gold standard SRS estimates suggests that HMIS fares well with slight discrepancies with reference to SRS. However, in the absence of other reliable data sources at the micro-level (district-level) in India, HMIS fills the gap with decent quality information that can help policy and planning at district level in the country.

## Discussion

Considering the global SDG targets, all countries are expected to have MMR below 70 per 100,000 live births, and no country with MMR above 140 per 100,000 live births by 2030 [29, 30]. The Government of India has also launched National Health Mission in 2015, subsuming the previous National Rural Health Mission and National Urban Health Mission to bring out the necessary structural changes in the public health care and delivery system in India. The National Health Mission design provides the Reproductive-Maternal-Neonatal-Child and Adolescent Health Services, strengthening the health system to achieve important demographic and health goals. Schemes like Janani Suraksha Yojana under National Rural Health Mission have contributed significantly to the rise in antenatal care and institutional deliveries, thereby reducing MMR [10–12]. However, there is no investigation regarding the programme success at the sub-national level, especially at the level of micro administrative units like districts.

In view of this, we make a significant contribution through this study. Our findings reveal that 71 percent of India's total districts (456 out of 640 districts) have reported MMR above 140. According to SRS (2016–18), only Assam (215) has MMR of more than 200, while our district-level assessment based on HMIS suggests that about 130 districts have reported above 200

maternal deaths per 100,000 live births. Thus, our mapping and spatial analyses findings highlight a greater spatial heterogeneity across districts in the country, with spatial clustering (hotspots) of high MMR in the North-eastern and Central regions, and low MMR in the Southern and Western regions. However, we have also observed considerable within-state variations in states across their districts. Even the better-off states such as Kerala, Tamil Nadu, Andhra Pradesh, Karnataka, and Gujarat have medium to high MMR pockets that need policy attention. Owing to data availability, only Assam from the North-eastern region was in the lime-light for higher maternal deaths, but with this study, it has been learned that the entire region is facing a similar problem and needs policy attention.

Our assessment of socioeconomic correlates of MMR suggests that improvement in antenatal care, postnatal care within 48 hours of delivery, body mass index, years of schooling, and reduction of higher-order births, births in higher ages, and poor economic status will help in reducing MMR in the districts of India. The districts with better health infrastructure have significantly less MMR, while those with a high SC/ST population show higher MMR levels. However, the most surprising factor is the lack of a significant negative association of institutional deliveries with MMR. Looking at this finding in conjunction with previous studies, which showed an unexpected relationship with both infant mortality and maternal mortality, suggests that it may be because a considerable number of women rush to institutional deliveries when complications arise; most often a majority of them have not obtained full and quality antenatal care. Thus, risky deliveries contribute to a greater number of deaths at the institutions compared to home deliveries [10, 11, 13, 31]. In particular, Randive and colleagues found a gap between access to just institutional deliveries and access to emergency obstetric care, perhaps demonstrating that women delivering in institutions are not automatically receiving sufficient care [13]. Another startling finding is the positive relationship between contraceptive use and MMR in one of the models. Such a relationship is possible in the context of low quality of care in family planning, leading to greater maternal morbidity and increasing the risk of obstetric complications and mortality [18]. However, in the full model, where all the covariates were controlled, the coefficient for contraceptive use becomes insignificant.

From a policy perspective, the study's findings advance two key messages: first, despite decent progress in reducing maternal mortality, several districts in India need to initiate immediate action to meet the ambitious SDG-3 target of MMR and ultimately eliminate preventable maternal mortality. The states that made a concerted effort to reduce maternal mortality, especially post-2005, provide pathways to accomplish the acceleration necessary to reduce preventable maternal deaths substantially. In particular, post-2005 MMR reduction in Maharashtra, Telangana, and Andhra Pradesh is very impressive [5, 11].

Secondly, the study highlights that maternal health care, especially postnatal care, and maternal nutrition are key for reducing maternal mortality. Considering that children ever born, years of schooling, and poor household economic status also emerged as critical factors; avoiding higher-order births, ensuring dissemination of right maternal health knowledge, and affordable essential services help in accelerating the process of reduction in MMR. Despite Janani Suraksha Yojana being in place, out of pocket expenditure on maternal health care in several states of India is way higher than Janani Suraksha Yojana incentives [32–34]; which might be impacting accessing quality antenatal and institutional delivery care and, as a result, this is impacting on reducing maternal mortality. Therefore, the ongoing Pradhan Mantri Matriva Vandhana Yojana must consider raising Janani Suraksha Yojana incentives to ensure affordable and quality maternal health care for all. Moreover, a significant association between sex ratio at birth and MMR suggests that maternal deaths are also happening due to unsafe abortions and thus need policy attention. A highly developed state like Punjab falling in the moderate to high MMR category also raises the question that unsafe sex-selective abortions

may be contributing to maternal deaths. However, considering that HMIS collects only facility-based data, the analysis of the determinants is essentially an ecological study, thus prone to ecological bias.

Third, although the reliability of routinely recorded mortality data by health system employees has been continuously questioned [35] if it is handled well, systems like HMIS would be a permanent solution to the long-standing problem of the absence of micro-level demographic and health information in India. Despite some caveats associated with HMIS data on maternal deaths, in the absence of any other reliable data sources at the micro-level (district-level) in India, it fills the gap with decent quality information that can help policy and planning at the district level in the country. In general, vital registration systems such as HMIS lack political priority in several states, thus leading to poor management, supervision, and underfunding. While an efficient death reporting system may be more complex and entail institutional arrangements across many governmental departments, they can be made to work subject to strong regional momentum and leadership. Given the encouraging results already achieved with minimal support for HMIS, an integrated review system and supervision should probably produce better results. SRS kind of double recording system (vital registration and survey-based confirmation) at least for one district in a state helps monitor, evaluate, and validate HMIS data. Therefore, our study will rejuvenate the plan of increasing efforts to revive the vital registration system at a national level with an inspiration of reasonably good quality registration evident in case of maternal deaths under HMIS.

## Supporting information

**S1 Fig. Bivariate LISA (cluster and significance) maps depicting spatial clustering and spatial outliers of maternal mortality ratio by selected background characteristics in India.**
(PDF)

**S1 Table. Strengthening the reporting of observational studies or cross-sectional studies in epidemiology (STROBE checklist).**
(PDF)

**S2 Table. State-wise estimates of MMR from SRS and HMIS.**
(PDF)

**S3 Table. District-wise estimates of MMR from HMIS.**
(PDF)

**S4 Table. Comparison of infant mortality rate, sex ratio at birth and crude birth rate from SRS and HMIS.**
(PDF)

## Acknowledgments

We thank P.M. Kulkarni (Rtd Professor, Center for the Studies in Regional Development, Jawaharlal Nehru University) for numerous discussions about HMIS data and methodology of MMR estimation using HMIS; and also, for reviewing the first draft of the paper. We also thank Prof. Christophe Guilmoto (a senior fellow in Demography at the French Institut de Recherche pour le Développement (IRD)) and Prof. Arvind Pandey (Former Director, National Institute of Medical Statistics, New Delhi) for useful discussion about the demographic estimates using the vital registration system data.

## Author Contributions

**Conceptualization:** Srinivas Goli, K. S. James.

**Data curation:** Parul Puri, Pradeep S. Salve.

**Formal analysis:** Srinivas Goli, Parul Puri.

**Investigation:** Saseendran Pallikadavath, K. S. James.

**Methodology:** Srinivas Goli, Parul Puri.

**Project administration:** Srinivas Goli, Saseendran Pallikadavath, K. S. James.

**Supervision:** Srinivas Goli, Saseendran Pallikadavath, K. S. James.

**Writing – original draft:** Srinivas Goli, Pradeep S. Salve.

**Writing – review & editing:** Srinivas Goli, Saseendran Pallikadavath, K. S. James.

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
