## [Decision Letter · Decision Letter 0]

22 Feb 2022

PGPH-D-22-00024

Estimates and Correlates of District-Level Maternal Mortality Ratio in India

Dear Dr. Goli,

Thank you for submitting your manuscript to PLOS Global Public Health. After careful consideration, we feel that it has merit but does not fully meet PLOS Global Public Health’s publication criteria as it currently stands. Therefore, we invite you to submit a revised version of the manuscript that addresses the points raised during the review process.

We look forward to receiving your revised manuscript.

Kind regards,

Stephen J. McCall, DPhil

Academic Editor

Journal Requirements:

1. Please amend your Financial Disclosure statement. If you did not receive any funding for this study, please simply state: “The authors received no specific funding for this work.”

2. Please update your Competing Interests statement. If you have no competing interests to declare, please state: “The authors have declared that no competing interests exist.”

3. Please provide us with a direct link to the base layer of the map used in Figures 1, 2, and 3, and ensure this location is also included in the figure legend. 

Please note that, because all PLOS articles are published under a CC BY license (creativecommons.org/licenses/by/4.0/), we cannot publish proprietary maps such as Google Maps, Mapquest or other copyrighted maps. If your map was obtained from a copyrighted source please amend the figure so that the base map used is from an openly available source.

Please note that only the following CC BY licences are compatible with PLOS licence: CC BY 4.0, CC BY 2.0  and CC BY 3.0, meanwhile such licences as CC BY-ND 3.0 and others are not compatible due to additional restrictions. If you are unsure whether you can use a map or not, please do reach out and we will be able to help you. 

The following websites are good examples of where you can source open access or public domain maps:

Additional Editor Comments (if provided):

Reviewers' comments:

Reviewer's Responses to Questions

**Comments to the Author**

1. Does this manuscript meet PLOS Global Public Health’s publication criteria? Is the manuscript technically sound, and do the data support the conclusions? The manuscript must describe methodologically and ethically rigorous research with conclusions that are appropriately drawn based on the data presented.

Reviewer #1: Partly

Reviewer #2: Yes

Reviewer #3: Yes

2. Has the statistical analysis been performed appropriately and rigorously?

Reviewer #1: No

Reviewer #2: Yes

Reviewer #3: No

3. Have the authors made all data underlying the findings in their manuscript fully available (please refer to the Data Availability Statement at the start of the manuscript PDF file)?

Reviewer #1: Yes

Reviewer #2: Yes

Reviewer #3: Yes

4. Is the manuscript presented in an intelligible fashion and written in standard English?

Reviewer #1: Yes

Reviewer #2: Yes

Reviewer #3: Yes

5. Review Comments to the Author

Reviewer #1: Reviewer’s comments

1. The table 2, requires appropriate titles for the columns. The column 'Obs' does not signify whether the value is for number of districts, or the the absolute number for each of the variable.

2. There are different factors operating for IMR & MMR. IMR as proxy measure for MMR does not suit.

3. Required to be mention in details, the method for district estimates were adjusted was done?

4. Mention the cut off values used for categorization of high or low MMR.

5. The uniform nomenclature to be maintained either states and UTs or provinces.

6. Is there any significance of the information provided in the lines 309 to 311, if not; it can be removed.

7. The validity and reliability check was done for the MMR. Similarly it may be done for factors associated with MMR.

Reviewer #2: Many thanks for asking me review this paper assessing the sub-district level maternal mortality ratio in India using HMIS data. This is a very useful analysis for monitoring progress towards achieving the SDG MMR target. As India is a very large country with health devolved to state governments it is important to obtain disaggregated state/ district level data to assess the within country variability. The analysis is appropriate and the manuscript is by and large well written. I have a few suggestions to improve the manuscript:

1. The introduction is still very long and would suggest some of that can be moved to the discussion section. For example, the paragraph lines 111 to 122 will sit better in the discussion section.

2. There is an assumption that reader is familiar with the sample registration system, HMIS etc. It will be useful to get a description of these data sources - particularly these two with a description of how the data are collected.

3. Some clarification regarding the limitations of SRS data and how HMIS can overcome these limitations would be good.

4. The methods section states that data triangulation was done using several different databases. However, table 1 does not include for example, census data. Please could you clarify how this data was used? On the same note, the link for data only appears to give HMIS data. Can the authors please confirm that it will be possible to repeat all the analyses reported in the paper using this data alone?

5. Please state the strengths and limitations of the study in the discussion section. In particular please mention that HMIS will only collect facility based data and that the determinants analysis is essentially an ecological study and therefore prone to ecological bias.

6. It will be good to report any monitoring of maternal deaths such as Confidential Enquiries into Maternal Deaths at national, state and health facility level. if not, future recommendation for automated HMIS data to feed into such enquiries can be suggested.

Reviewer #3: A very interesting study. I applaud the authors for their hard work and analysis. I recommend publication if my comments can be addressed in the subsequent revision.

Major comments:

Epidemiological studies are moving away from presenting models that adjust for all variables in the same model (although it is still very prevalent in the literature) unless these variables are theoretically underpinned as this can lead to erroneous results. For example, poor household economic status changed direction with MM after adjusting for other variables. Does this mean that poor household economic status is not correlated or that other variables explain the relationship between poor HH economic status and MM? To avoid this error the analysis needs to present (1) univariable analysis (2) multivariable models for each "MM vs. factor relationship" adjusting only for variables that can be confounders, ideally informed by causal diagrams.

References:

Westreich, Daniel, and Sander Greenland. "The table 2 fallacy: presenting and interpreting confounder and modifier coefficients." American journal of epidemiology 177.4 (2013): 292-298.

Griffith, Gareth J., et al. "Collider bias undermines our understanding of COVID-19 disease risk and severity." Nature communications 11.1 (2020): 1-12. It is likely more appropriate to present each variables in a univariable model rather than presenting a fully adjusted model.

Laubach, Zachary M., et al. "EIC (Expert Information Criterion) not AIC: the cautious biologist's guide to model selection." arXiv preprint arXiv:2010.07506 (2020).

Tennant, Peter WG, et al. "Use of directed acyclic graphs (DAGs) to identify confounders in applied health research: review and recommendations." International journal of epidemiology 50.2 (2021): 620-632.

Can you specify in the manuscript the proportion of the population where IMR was used as a proxy? Can the justification of the use of IMR be further expanded in the methods section.

Minor comments:

Please can the introduction be reduced and the limitations of the data sources be expanded.

Lots of abbreviations in the manuscript - please can you reduce these to make the paper more readable.

Tables - make sure they can be read without referring to the paper (make sure abbreviations are all explained in the footnotes)

6. PLOS authors have the option to publish the peer review history of their article (what does this mean?). If published, this will include your full peer review and any attached files.

**Do you want your identity to be public for this peer review?** For information about this choice, including consent withdrawal, please see our Privacy Policy.

Reviewer #1: **Yes: **Dr Archana Rathod

Reviewer #2: **Yes: **Dr Sohinee Bhattacharya

Reviewer #3: No

---

## [Decision Letter · Decision Letter 1]

27 Jun 2022

Estimates and Correlates of District-Level Maternal Mortality Ratio in India

PGPH-D-22-00024R1

Dear Dr. Goli,

We are pleased to inform you that your manuscript 'Estimates and Correlates of District-Level Maternal Mortality Ratio in India' has been provisionally accepted for publication in PLOS Global Public Health.

Best regards,

Abraham D. Flaxman, Ph.D.

Academic Editor

Reviewer Comments (if any, and for reference):

Reviewer's Responses to Questions

**Comments to the Author**

1. If the authors have adequately addressed your comments raised in a previous round of review and you feel that this manuscript is now acceptable for publication, you may indicate that here to bypass the “Comments to the Author” section, enter your conflict of interest statement in the “Confidential to Editor” section, and submit your "Accept" recommendation.

Reviewer #2: All comments have been addressed

Reviewer #3: (No Response)

2. Does this manuscript meet PLOS Global Public Health’s publication criteria? Is the manuscript technically sound, and do the data support the conclusions? The manuscript must describe methodologically and ethically rigorous research with conclusions that are appropriately drawn based on the data presented.

Reviewer #2: Yes

Reviewer #3: Yes

3. Has the statistical analysis been performed appropriately and rigorously?

Reviewer #2: Yes

Reviewer #3: Yes

4. Have the authors made all data underlying the findings in their manuscript fully available (please refer to the Data Availability Statement at the start of the manuscript PDF file)?

Reviewer #2: Yes

Reviewer #3: Yes

5. Is the manuscript presented in an intelligible fashion and written in standard English?

Reviewer #2: Yes

Reviewer #3: Yes

6. Review Comments to the Author

Reviewer #2: All my comments and suggestions have been addressed appropriately.

Reviewer #3: Thank you for addressing my comments.

One final comment:

Any two variables that have a correlation >0.3 I would consider highly correlated. Please consider separating these from the model. ANCs, PNCs, and Institutional deliveries.

7. PLOS authors have the option to publish the peer review history of their article (what does this mean?). If published, this will include your full peer review and any attached files.

**Do you want your identity to be public for this peer review?** For information about this choice, including consent withdrawal, please see our Privacy Policy.

Reviewer #2: **Yes: **Dr Sohinee Bhattacharya

Reviewer #3: No
